# Clinical Observation of SGLT2 Inhibitor Therapy for Cardiac Arrhythmia and Related Cardiovascular Disease in Diabetic Patients with Controlled Hypertension

**DOI:** 10.3390/jpm12020271

**Published:** 2022-02-12

**Authors:** Shih-Jie Jhuo, Tsung-Hsien Lin, Yi-Hsiung Lin, Wei-Chung Tsai, I-Hsin Liu, Bin-Nan Wu, Kun-Tai Lee, Wen-Ter Lai

**Affiliations:** 1Graduate Institute of Clinical Medicine, College of Medicine, Kaohsiung Medical University, Kaohsiung 80756, Taiwan; jhuoshihjie@gmail.com (S.-J.J.); k920265@gap.kmu.edu.tw (W.-C.T.); kuntai.lee@yahoo.com.tw (K.-T.L.); 2Department of Internal Medicine, Division of Cardiology, Kaohsiung Medical University Hospital, Kaohsiung 80756, Taiwan; lth@kmu.edu.tw (T.-H.L.); caminolin@gmail.com (Y.-H.L.); ihsin343@yahoo.com.tw (I.-H.L.); wtlai@kmu.edu.tw (W.-T.L.); 3Department of Internal Medicine, Department of Pharmacology, Faculty of Medicine, College of Medicine, Kaohsiung Medical University, Kaohsiung 80756, Taiwan; 4Lipid Science and Aging Research Center, Kaohsiung Medical University, Kaohsiung 80756, Taiwan; 5Center for Lipid Biosciences, Kaohsiung Medical University Hospital, Kaohsiung 80756, Taiwan

**Keywords:** SGLT2 inhibitor, diabetes, atrial fibrillation, cardiac arrhythmia

## Abstract

Sodium-glucose transporter 2 (SGLT2) inhibitors are new glucose-lowering agents that have been proven to be beneficial for patients with cardiovascular diseases, heart failure, and sudden cardiac death. However, the possible protective effects of cardiac arrhythmia have not yet been clarified in clinical practice. In this study, we attempted to demonstrate the effects of SGLT2 inhibitors on cardiac arrhythmia by medical records from a single center. This retrospective study included patients diagnosed with type 2 diabetes mellitus (DM) and controlled hypertension who prescribed the indicated glucose-lowering agents based on medical records from 2016 to 2019 from Kaohsiung Medical University Hospital. These patients were divided into two groups. Group one patients were defined as patients with SGLT2 inhibitor therapy, and group two patients were defined as patients without SGLT2 inhibitor therapy. Baseline characteristics were collected from medical records. Univariate, multivariate, and match-paired statistical analyses were performed for the study endpoints. The primary study outcome was the incidence of cardiac arrhythmias, including atrial and ventricular arrhythmias, after SGLT2 inhibitor therapy. The secondary study outcomes were the incidence of stroke, heart failure, and myocardial infarction after SGLT2 inhibitor therapy. From the initial 62,704 medical records, a total of 9609 people who met our experimental design criteria were included. The mean follow-up period was 51.50 ± 4.23 months. Group one included 3203 patients who received SGLT2 inhibitors for treatment, and group two included 6406 patients who received non-SGLT2 inhibitors for treatment. Multivariate analysis showed that group one patients had significantly lower incidences of total cardiac arrhythmia (hazard ratio (HR): 0.58, 95% confidence interval (CI): 0.38–0.89, *p* = 0.013) and atrial fibrillation (HR: 0.56, 95% CI: 0.35–0.88, *p* = 0.013) than those of group two patients. The secondary outcome analysis showed that group one patients also had a significantly lower risk of stroke (HR: 0.48, 95% CI: 0.33–0.7; *p* < 0.001), heart failure (HR: 0.54, 95% CI: 0.41–0.7, *p* < 0.001), and myocardial infarction (HR: 0.47, 95% CI: 0.31–0.72, *p* < 0.001). A time-to-event analysis showed that treatment of type 2 DM patients with SGLT2 inhibitors could reduce the probability of total cardiac arrhythmia and related cardiovascular disease, such as atrial fibrillation, stroke, heart failure, or myocardial infarction, by 0.5%~0.8%. This databank analysis showed that SGLT2 inhibitor therapy reduced the incidence of total cardiac arrhythmia and atrial fibrillation in type 2 DM patients and decreased the incidence of related cardiovascular diseases, such as stroke, heart failure, and myocardial infarction. However, additional investigations are needed to confirm this hypothesis.

## 1. Introduction

Type 2 diabetes mellitus (DM) is a well-known risk factor for cardiovascular disease [1,2], and previous studies have indicated that it could also modify the electrophysiological characteristics of arrhythmic substrates and may be considered a risk factor for cardiac arrhythmia [3,4]. Traditional glucose-lowering agents, such as sulfonylurea, have not shown significant antiarrhythmic effects in clinical practice [5,6]. Although sulfonylurea reduced myocardial fibrosis, it could also cause electrocardiographic QT interval prolongation, which results in cardiac arrhythmia [7].

Sodium-glucose transporter 2 (SGLT2) inhibitors are a new generation of glucose-lowering agents that inhibit glucose reabsorption in renal tubules [8]. In the EMPA-REG trial, type 2 DM patients at high risk for cardiovascular events treated with SGLT2 inhibitors had significantly decreased cardiovascular mortality and heart failure hospitalizations [9]. Patients with heart failure who presented an ejection fraction of the left ventricle of 40% or less also had a significantly lower incidence of death from cardiovascular causes and hospitalization after SGLT2 inhibitor therapy in the DAPA-HF trial [10]. Several reports have shown that SGLT2 inhibitors do not prolong the QT interval on electrocardiography, alleviate atrial remodeling, or improve mitochondrial function in diabetic rats [11,12]. In our previous reports, the ion currents of cardiomyocytes could be modulated by adipocytokines from fat tissues after SGLT2 inhibitor therapy to reduce arrhythmogenesis [13]. SGLT2 inhibitor therapy also reduced ventricular fibrosis and connexin downregulation, which implied lower arrhythmogenicity [14]. Although SGLT2 inhibitors were reported to decrease the risk of atrial fibrillation, clinical practice data were not available to demonstrate and confirm the antiarrhythmic effect of SGLT2 inhibitor therapy [15]. Therefore, this study was performed to clarify the effects of SGLT2 inhibitor therapy on cardiac arrhythmia in clinical practice.

## 2. Material and Methods

### 2.1. Study Population

This retrospective study included patients with type 2 DM and indicated glucose-lowering agents from medical records from Kaohsiung Medical University Hospital from 2016 to 2019. Patients who did not use glucose-lowering agents, had an age of less than 18 years, and those with uncontrolled hypertension were excluded (blood pressure >140/90 mmHg). Included patients were taking glucose-lowering agents for at least 3 months. In addition, patients with a history of arrhythmia and anti-arrhythmic medication use were also excluded. Patients with cardiac arrhythmia were defined by International Classification of Diseases (ICD) 9/10 coding in medical records, which includes atrial and ventricular arrhythmia. The anti-arrhythmic medication included dronedarone, propafenone, amiodarone, flecainide, and mexiletine. The characteristics of the included patients were collected for univariate and multivariate analysis by medical records. The included patients were divided into 2 groups. Group 1 was defined as patients receiving SGLT2 inhibitor therapy, and group 2 was defined as patients not receiving SGLT2 inhibitor therapy. After collecting the baseline characteristics, the association and difference in SGLT2 inhibitor therapy and cardiac arrhythmia between the two groups were investigated. This study was approved by the Institutional Review Board of Kaohsiung Medical University Hospital (KMUHIRB-E(II)-20180143).

### 2.2. Study Outcomes

In this study, univariate and multivariate analyses were performed to investigate study outcomes. The primary study outcome was the incidence of atrial and ventricular arrhythmias and their association with SGLT2 inhibitor therapy. The secondary study outcome was a compound of related cardiovascular events including stroke, heart failure, and myocardial infarction and their association with SGLT2 inhibitor therapy All outcome events were defined by ICD9/10 coding in medical records during the following period.

### 2.3. Statistical Analysis

Descriptive data were expressed as counts, percentages of categorical variables, and the mean ± standard deviation (SD) of continuous variables. χ^2^ and Fisher’s exact test were used for discrete variables, and Student’s t test or the Mann–Whitney U test was used for comparison of continuous variables. We first conducted a univariate analysis and then established multivariate models by including variates that showed significance in the univariate model. Subsequently, multivariate and match-paired analysis were performed to identify the association of study variables and SGLT inhibitor therapy. Univariate, multivariate, and match-paired predictors used Cox regression to calculate the hazard ratio and *p*-value. A Kaplan–Meier plot was used to estimate the event incidence after treatment with or without SGLT2 inhibitors. The important parameters selected from the baseline characteristics or the clinical correlation with the study endpoints were used to achieve significance (*p* < 0.05 in univariate analysis). Subsequently, multivariate and match-paired Cox regression analysis were performed. *p* < 0.05 was defined as significant. The relationship between individual factors and the endpoint was expressed as a hazard ratio (HR) with a 95% confidence interval (CI). All analyses were performed using SPSS version 20 (SPSS Inc., Chicago, IL, USA).

## 3. Results

### 3.1. Characteristics of Included Patients

In this study, a total of 62,704 patients were initially included in our database. A total of 23,403 (37.3%) patients had type 2 DM and controlled hypertension, and were administered glucose-lowering agents. Among the 23,403 people, 516 patients were excluded because of symptoms and records of arrhythmia before inclusion, and 9311 patients were excluded because of incomplete clinical data for tracking and counting. Finally, 13,576 patients were included in our study. In addition, we also conducted an age- and sex-matched analysis of patients with SGLT2 inhibitors, and the numbers of matched group one patients and group two patients were 3203 and 6406 cases, respectively. The mean follow-up period was 51.50 ± 4.23 months. The inclusion flow chart is presented in Figure 1.

The baseline characteristics of the included patients are listed in Table 1. The mean follow-up period was 51.50 ± 4.23 months. Through the matched-pairs multivariate analysis, there were no significant differences between age, sex, and GPT in the two groups. The average age of included patients was about 65 years with 58–59% being male. However, higher values of low-density lipoprotein-cholesterol, total cholesterol, and triglyceride were found in group two patients. Group one patients had slightly higher values of fasting plasma glucose (GLU), high-density lipoprotein-cholesterol (HDL-C), and HbA1c than group two patients.

### 3.2. Association between Total Cardiac Arrhythmia Incidence and SGLT2 Inhibitor Therapy

A Cox regression model was used to investigate the association of these risk factors with the incidence of total cardiac arrhythmia in the presence of SGLT2 inhibitor therapy, and the results are listed in Table 2. In the included patients, SGLT2 inhibitor therapy, age, GPT, LDL-C, T-CHO, HDL-C, and gout were significant factors for total cardiac arrhythmia incidence in the univariate analysis. The age- and sex-matched multivariate analysis showed that SGLT2 inhibitor therapy significantly reduced the risk of total cardiac arrhythmia (HR: 0.58, 95% CI: 0.38–0.89, *p* = 0.013). In addition, HDL-C corresponded to a significantly decreased incidence of total cardiac arrhythmia (HR: 0.73, 95% CI: 0.58–0.91, *p* = 0.005).

### 3.3. Association between Atrial Fibrillation Incidence and SGLT2 Inhibitor Therapy

The association of AF incidence with SGLT2 inhibitor therapy was investigated by Cox regression, and the results are listed in Table 3. Similar to the results for total arrhythmia, SGLT2 inhibitor therapy, age, GPT, LDL-C, T-CHO, HDL-C, and gout were significant factors for AF incidence in the univariate analysis. The matched multivariate analysis indicated that SGLT2 inhibitor therapy significantly reduced the incidence of atrial fibrillation (HR: 0.56, 95% CI: 0.35–0.88, *p* = 0.013). In addition, the value of T-CHO was also significantly negatively correlated with the incidence of atrial fibrillation (HR: 0.64, 95% CI: 0.41–1.00, *p* = 0.048) in our study cohort (Table 3).

### 3.4. Association between Stroke and SGLT2 Inhibitor Therapy

In the secondary outcome analysis, we investigated the association of SGLT2 inhibitor therapy with three related cardiovascular diseases, including stroke, heart failure, and myocardial infarction. Group one patients had lower incidences of stroke (group one vs. group 2: 1.0% vs. 2.1%), heart failure (2.1% vs. 3.7%), and myocardial infarction (0.9% vs. 1.5%) than those of group two patients. In the univariate analysis, SGLT2 inhibitor therapy, age, HDL-C, and gout were significant factors for stroke incidence. The age- and sex-matched multivariate analysis were also conducted, which showed that SGLT2 inhibitor therapy (HR: 0.48, 95% CI: 0.33–0.7; *p* < 0.001) and HDL-C (HR: 0.61, 95% CI: 0.51–0.74, *p* < 0.001) significantly reduced the incidence of stroke in our cohort (Table 4).

### 3.5. Association between Heart Failure and SGLT2 Inhibitor Therapy

Table 5 shows the results of the univariate and multivariate analyses between SGLT2 inhibitor therapy and heart failure. Univariate analysis showed that SGLT2 inhibitor therapy, age, GPT, LDL-C, T-CHO, HDL-C, gout, and HbA1c were significant factors for heart failure. The matched multivariate analysis of the Cox regression results showed that SGLT2 inhibitor therapy (HR: 0.54, 95% CI: 0.41–0.7, *p* < 0.001) and HDL-C (HR: 0.78, 95% CI: 0.68–0.89, *p* < 0.001) could significantly reduce the incidence of heart failure. However, HbA1C value (HR: 1.18, 95% CI: 1.18–1.31, *p* = 0.001) was found to be a risk factor for heart failure.

### 3.6. Association between Myocardial Infarction and SGLT2 Inhibitor Therapy

In this study, we also evaluated the correlation between myocardial infarction and SGLT2 inhibitor therapy. Table 6 reveals the univariate analysis showing that SGLT2 inhibitor therapy, age, male sex, GPT, LDL-C, T-CHO, and HDL-C were significant factors for myocardial infarction. The matched multivariate analysis showed that SGLT2 inhibitor therapy significantly reduced the incidence of myocardial infarction (HR: 0.47, 95% CI: 0.31–0.72, *p* < 0.001). In addition, HDL-C (HR: 0.63, 95% CI: 0.5–0.79, *p* < 0.001) significantly reduced the incidence of myocardial infarction.

### 3.7. Kaplan–Meier Plot for the Time-to-Event Incidence of Cardiac Arrhythmia and Related Cardiovascular Disease in Type 2 DM Patients

Figure 2 shows the Kaplan–Meier plots for the time-to-event incidence of the primary and secondary study outcomes. The results showed that compared with group two patients, the incidence of total cardiac arrhythmia in group one patients was significantly reduced by 0.6%. The findings also determined that group one patients had a 0.7% lower risk of atrial fibrillation than that of group two patients. In the secondary study outcome analysis, group one patients had a 0.5% lower incidence rate of stroke than that of group two patients. Moreover, the incidence of heart failure and myocardial infarction in group one patients were lower than that in group two patients, both significantly reduced by 0.8%. These results showed that group one patients had a lower risk of total cardiac arrhythmia, atrial fibrillation, and related cardiovascular diseases, such as stroke, heart failure, and myocardial infarction.

## 4. Discussion

### 4.1. Main Findings

There were three main findings in our study. First, compared with patients treated with other glucose-lowering agents, patients treated with SGLT2 inhibitors had a lower incidence of cardiac arrhythmia in clinical practice. Second, the incidence of atrial fibrillation was also lower in patients treated with SGLT2 inhibitors. Third, the incidence of related cardiovascular diseases, such as stroke, heart failure and myocardial infarction, was lower in patients treated with SGLT2 inhibitors than in those treated with other glucose-lowering agents.

### 4.2. Association between SGLT2 Inhibitor Treatment and Cardiac Arrhythmia

Our study showed that the administration of SGLT2 inhibitors reduced the incidence of cardiac arrhythmia in type 2 DM patients. Metabolic syndrome and type 2 DM are well-known risk factors for cardiac arrhythmia. Several possible hypotheses have been extensively discussed, including modification of the autonomic system, oxidative processes, and inflammatory reactions [16]. In our previous reports, we also found that metabolic syndrome could not only decrease the delayed-rectified potassium current (I_K_) of cardiomyocytes but could also increase L-type calcium channel current (I_Ca, L_) overload, which might cause arrhythmogenesis [13]. At the same time, it also decreased the expression of connexin and increased the fibrotic area of ventricular tissues [14]. After SGLT2 inhibitor therapy, the I_K_ decreased, and I_Ca, L_ overload improved. At the same time, the decreases in connexin expression and the fibrosis area were also attenuated. These findings might indicate the possible mechanisms of SGLT2 inhibitors in cardiac arrhythmia. In clinical practice, a population-based cohort study revealed that type 2 DM patients treated with SGLT2 inhibitor therapy had a lower risk of new-onset arrhythmia and mortality than those of patients treated with non-SGLT2 inhibitor therapy [17]. For type 2 DM patients with myocardial infarction, SGLT2 inhibitor therapy might modulate the cardiac autonomic nervous system and result in fewer adverse events [18]. These findings were compatible with those of our study. A clinical study to assess the effect of SGLT2 inhibitors on ventricular arrhythmia in type 2 DM patients with intracardiac defibrillators is also in progress, and more evidence is needed to consolidate the findings.

### 4.3. Association between SGLT2 Inhibitor Treatment and Atrial Fibrillation

Atrial fibrillation is the most prevalent cardiac arrhythmia in the world, and it can increase the incidence of ischemic stroke, cardiovascular disease, and heart failure [19]. There are many risk factors for atrial fibrillation, including metabolic syndrome and diabetes. For patients with type 2 DM, increased visceral adipose tissue, autonomic dysfunction, and atrial fibrosis were found [3], which might result in atrial fibrillation. However, after traditional glucose-lowering agent therapy, the incidence of atrial fibrillation was not significantly decreased in clinical practice. In the DECLARE-TIMI 58 trial, the subgroup analysis showed that dapagliflozin decreased the incidence of atrial arrhythmia in patients with type 2 DM, regardless of a previous history of atrial fibrillation, cardiovascular disease, or heart failure [15]. Similar results were also shown in other meta-analyses and studies [20,21]. Several possible hypotheses have been extensively discussed, including reductions in body weight, blood pressure, and heart failure incidence [20]. In previous studies, an SGLT2 inhibitor was shown to decrease the amount of pericardial fat and atrial fibrosis [22,23], which might attenuate the occurrence of atrial fibrillation. Several inflammatory cytokines were also revealed to be modified after SGLT2 inhibitor therapy [24,25], such as the NLRP3 inflammasome, which might reduce the area of cardiac fibrosis. In this study, we observed that the incidence of atrial fibrillation was significantly lower in patients treated with SGLT2 inhibitors than in those treated with other glucose-lowering agents in clinical practice. This study provides clinical practice evidence that SGLT2 inhibitors decrease the incidence of atrial fibrillation and have other related cardioprotective effects.

### 4.4. Association between SGLT2 Inhibitor Treatment and Other Cardiovascular Outcomes

Type 2 DM is also a well-known risk factor for cerebrovascular and cardiovascular disease. Some glucose-lowering agents were previously reported to decrease the risk and severity of ischemic stroke and myocardial infarction [26,27,28]. In our study, clinical practice evidence demonstrated a lower incidence of ischemic stroke and myocardial infarction after SGLT2 inhibitor therapy. There are many possible mechanisms that can explain these results, including better control of blood pressure or heart failure. For ischemic stroke and myocardial infarction, atherosclerosis of the artery is an important risk factor, and the amount of visceral fat has a significant association with atherosclerosis [29,30]. In previous studies, SGLT2 inhibitor therapy was proven to significantly decrease the amount of visceral fat [22,31]. In addition, SGLT2 inhibitors might also reduce adipose-mediated inflammation and proinflammatory cytokine production [32]. Both effects might result in a reduction in atherosclerosis. Beyond the reduction in cardiovascular and cerebrovascular events, SGLT2 inhibitors also decrease the incidence of and hospitalization for heart failure [9,10]. Several possible mechanisms have been extensively discussed, including blood pressure reduction, cardiac metabolism modification, ion channels, and cardiac fibrosis reduction [33,34]. Therefore, SGLT2 inhibitors have become the fifth pillar of heart failure management [35]. Similar effects were also observed in our study, thus providing clinical evidence of SGLT2 inhibitor therapy in heart failure management.

## 5. Limitations

In this study, there were several limitations. First, the study was retrospective and had inherent limitations. We investigated the relationship based on recorded clinical information, which might introduce bias. Second, the medical records of patients were collected for only 4 years, and there is information loss, such as of BMI value and BP, in our databank. We will include more clinical data to make our database more complete. Additional medical records and longer study times might be necessary to clarify the differences. Third, the clinical examination of choice are risk factors for arrhythmia, atrial fibrillation, or other cardiac disease, the potential bias of which warrants further investigation. Finally, a unified protocol for cardiac arrhythmia detection was not observed in the medical records. Therefore, the possibility of undetected arrhythmia cannot be ruled out in this study.

## 6. Conclusions

Our study showed that SGLT2 inhibitor therapy has beneficial effects on cardiac arrhythmia and reduces the incidence of atrial fibrillation in clinical practice. In addition, this therapy also reduces the incidence of stroke, heart failure, and myocardial infarction. Although several possible mechanisms were discussed, additional investigations are required to confirm these findings.

## Figures and Tables

**Figure 1 jpm-12-00271-f001:**
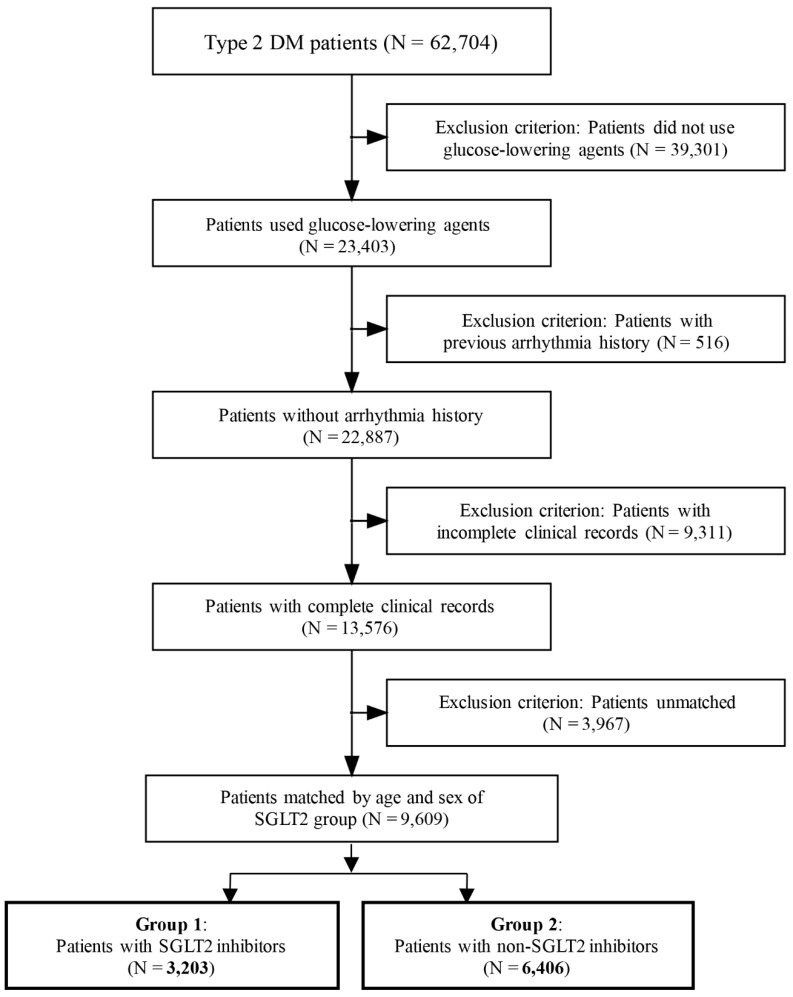
Flowchart of patients included in the current study and inclusion and exclusion criteria for the study proposal. Patients with SGLT2 inhibitor treatment were grouped into group 1, and those without SGLT2 inhibitors were included into group 2.

**Figure 2 jpm-12-00271-f002:**
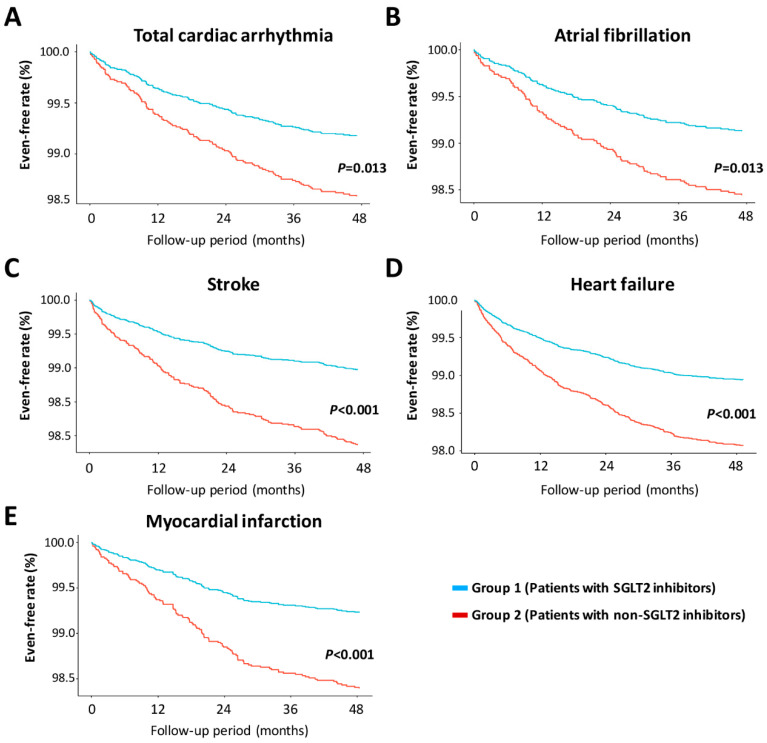
Kaplan–Meier plots demonstrated the time-to-event incidence of the primary and secondary study outcomes, including (**A**) total arrhythmia, (**B**) atrial fibrillation, (**C**) stroke, (**D**) heart failure, and (**E**) myocardial infarction. A significantly lower risk of cardiac arrhythmia and related cardiovascular disease was found in group 1 patients than in group 2 patients. All time-to-event incidence curves were generated after the multivariate Cox regression analysis.

**Table 1 jpm-12-00271-t001:** Characteristics of the patients included in the study.

Characteristic (Mean (SD))	Group 1 SGLT2, N = 3203	Group 2 non-SGLT2, N = 6406	*p*-Value
Age, years	65.333 (12.128)	65.628 (11.712)	0.6
Male sex, n (%)	1867/3203 (58%)	3757/6406 (59%)	0.7
GPT	28.151 (15.982)	29.808 (21.543)	0.2
HbA1c	7.513 (1.217)	7.139 (1.178)	<0.001
LDL-C	88.800 (26.372)	92.003 (26.190)	<0.001
T-CHO	158.809 (33.749)	163.341 (33.105)	<0.001
GLU (AC)	142.228 (36.529)	140.211 (36.173)	0.007
HDL-C	44.694 (11.512)	43.298 (11.648)	<0.001
TG	145.235 (132.144)	145.690 (100.912)	<0.001
Gout	2/3203 (<0.1%)	13/6406 (0.2%)	0.2

GLU (AC): fasting plasma glucose; GPT: glutamic pyruvate transaminase; Hb: hemoglobin; HDL-C: high-density lipoprotein-cholesterol; LDL-C: low-density lipoprotein-cholesterol; SD: standard deviation; T-CHO: total cholesterol; TG: triglyceride.

**Table 2 jpm-12-00271-t002:** Univariate and multivariate analyses of total cardiac arrhythmia incidence in the study patients.

Variable	Univariate Analysis			Multivariate Analysis			Matched-Pairs Analysis		
	HR	95% CI	*p*-Value	HR	95% CI	*p*-Value	HR	95% CI	*p*-Value
SGLT2 inhibitors therapy	0.5	(0.33–0.74)	0.001	0.61	(0.41–0.92)	**0.019**	0.58	(0.38–0.89)	**0.013**
Age	2.3	(1.9–2.7)	<0.001	1.95	(1.64–2.31)	**<** **0.** **001**			
Male	1.1	(0.85–1.5)	0.450						
GPT	0.74	(0.61–0.91)	0.005						
LDL-C	0.81	(0.7–0.94)	0.005	1.64	(1.20–2.26)	**0.002**			
T-CHO	0.7	(0.6–0.82)	<0.001	0.54	(0.38–0.77)	**<** **0.** **001**			
GLU(AC)	0.92	(0.8–1.1)	0.270						
HDL-C	0.71	(0.61–0.83)	<0.001				0.73	(0.58–0.91)	**0.005**
Gout	5.9	(1.5–24)	0.012						
HbA1C	1	(0.87–1.2)	0.66						

CI: confidence interval; GLU (AC): fasting plasma glucose, GPT: glutamic pyruvate transaminase; Hb: hemoglobin; HDL-C: high-density lipoprotein-cholesterol; HR: hazard ratio; LDL-C: low-density lipoprotein-cholesterol, T-CHO: total cholesterol; TG: triglyceride.

**Table 3 jpm-12-00271-t003:** Univariate and multivariate analyses of atrial fibrillation incidence in the study patients.

Variable	Univariate Analysis			Multivariate Analysis			Matched-Pairs Analysis		
	HR	95% CI	*p*-Value	HR	95% CI	*p*-Value	HR	95% CI	*p*-Value
SGLT2 inhibitors therapy	0.48	(0.31–0.73)	0.001	0.6	(0.39–0.94)	**0.025**	0.56	(0.35–0.88)	**0.013**
Age	2.4	(2.1–2.9)	<0.001	2.05	(1.71–2.47)	**<** **0.** **001**			
Male	1	(0.76–1.4)	0.9						
GPT	0.64	(0.5–0.82)	<0.001						
LDL-C	0.79	(0.68–0.93)	0.004	1.59	(1.13–2.24)	**0.008**			
T-CHO	0.7	(0.59–0.82)	<0.0001	0.55	(0.38–0.80)	**0.002**	0.64	(0.41–1.00)	**0.048**
GLU(AC)	0.89	(0.76–1)	0.150						
HDL-C	0.74	(0.63–0.87)	<0.001						
Gout	6.7	(1.7–27)	0.008						
HbA1C	1.1	(0.88–1.3)	0.55						

CI: confidence interval; GLU (AC): fasting plasma glucose, GPT: glutamic pyruvate transaminase; Hb: hemoglobin; HDL-C: high-density lipoprotein-cholesterol; HR: hazard ratio; LDL-C: low-density lipoprotein-cholesterol, T-CHO: total cholesterol; TG: triglyceride.

**Table 4 jpm-12-00271-t004:** Univariate and multivariate analyses for stroke incidence in the study patients.

Variable	Univariate Analysis			Multivariate Analysis			Matched-Pairs Analysis		
	HR	95% CI	*p*-Value	HR	95% CI	*p*-Value	HR	95% CI	*p*-Value
SGLT2 inhibitors therapy	0.41	(0.29–0.59)	<0.0001	0.51	(0.35–0.73)	**<0.001**	0.48	(0.33–0.7)	**<0.001**
Age	1.6	(1.4–1.8)	<0.0001	1.3	(1.14–1.47)	**<0.001**			
Male	1.1	(0.88–1.4)	0.39						
GPT	0.9	(0.78–1)	0.13						
LDL-C	1.1	(0.96–1.2)	0.23						
T-CHO	1	(0.93–1.2)	0.46						
GLU(AC)	1.1	(0.95–1.2)	0.31						
HDL-C	0.66	(0.58–0.75)	<0.0001	0.72	(0.63–0.82)	**<0.001**	0.61	(0.51–0.74)	**<0.001**
Gout	4.2	(1.1–17)	0.042						
HbA1C	1	(0.87–1.2)	0.86						

CI: confidence interval; GLU (AC): fasting plasma glucose, GPT: glutamic pyruvate transaminase; Hb: hemoglobin; HDL-C: high-density lipoprotein-cholesterol; HR: hazard ratio; LDL-C: low-density lipoprotein-cholesterol, T-CHO: total cholesterol; TG: triglyceride.

**Table 5 jpm-12-00271-t005:** Univariate and multivariate analyses of heart failure incidence in the study patients.

Variable	Univariate Analysis			Multivariate Analysis			Matched-Pairs Analysis		
	HR	95% CI	*p*-Value	HR	95% CI	*p*-Value	HR	95% CI	*p*-Value
SGLT2 inhibitors therapy	0.51	(0.4–0.64)	<0.001	0.6	(0.47–0.77)	**<** **0.001**	0.54	(0.41–0.7)	**<0.001**
Age	1.9	(1.7–2.1)	<0.001	1.57	(1.41–1.74)	**<** **0.001**			
Male	0.85	(0.72–1)	0.063						
GPT	66	(0.57–0.76)	<0.001	0.82	(0.72–0.94)	**0.003**			
LDL-C	0.78	(0.71–0.85)	<0.001	0.79	(0.68–0.92)	**0.002**			
T-CHO	0.85	(0.77–0.93)	<0.001	1.24	(1.08–1.43)	**0.003**			
GLU(AC)	0.99	(0.91–1.1)	0.86						
HDL-C	0.68	(0.62–0.75)	<0.001	0.77	(0.70–0.84)	**<** **0.001**	0.78	(0.68–0.89)	**<0.001**
Gout	8.6	(4.1–18)	<0.001						
HbA1C	1.1	(1–1.3)	0.019				1.18	(1.07–1.31)	**0.001**

CI: confidence interval; GLU (AC): fasting plasma glucose, Hb: hemoglobin; GPT: glutamic pyruvate transaminase; HDL-C: high-density lipoprotein-cholesterol; HR: hazard ratio; LDL-C: low-density lipoprotein-cholesterol, T-CHO: total cholesterol; TG: triglyceride.

**Table 6 jpm-12-00271-t006:** Univariate and multivariate analyses of myocardial infarction incidence in the study patients.

Variable	Univariate Analysis			Multivariate Analysis			Matched-Pairs Analysis		
	HR	95% CI	*p*-Value	HR	95% CI	*p*-Value	HR	95% CI	*p*-Value
SGLT2 inhibitors therapy	0.5	(0.34–0.74)	<0.001	0.47	0.31–0.70	<0.001	0.47	(0.31–0.72)	<0.001
Age	1.7	(1.5–1.9)	<0.001	1.46	1.23–1.72	<0.001			
Male	1.5	(1.2–2)	0.0025	1.68	1.24–2.26	<0.001			
GPT	0.66	(0.53–0.83)	<0.001	0.76	0.61, 0.95	0.018			
LDL-C	0.62	(0.53–0.73)	<0.001						
T-CHO	0.64	(0.55–0.74)	<0.001						
GLU(AC)	0.96	(0.84–1.1)	0.58						
HDL-C	0.56	(0.48–0.66)	<0.001	0.71	0.60, 0.85	<0.001	0.63	(0.5–0.79)	<0.001
Gout	<0.0001	(0-Inf)	0.99						
HbA1C	1.2	(0.98–1.4)	0.077						

CI: confidence interval; GLU (AC): fasting plasma glucose, GPT: glutamic pyruvate transaminase; Hb: hemoglobin; HDL-C: high-density lipoprotein-cholesterol; HR: hazard ratio; LDL-C: low-density lipoprotein-cholesterol, T-CHO: total cholesterol; TG: triglyceride.

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
