# Peer review of "Clinical Observation of SGLT2 Inhibitor Therapy for Cardiac Arrhythmia and Related Cardiovascular Disease in Diabetic Patients with Controlled Hypertension"

_jpm, 2022, doi:10.3390/jpm12020271_

Round 1

Reviewer 1 Report

SGLT2 Inhibitors and Arrhythmia

The authors present a retrospective cohort study on the association between SGLT2 therapy in patients with T2DM and onset of cardiac arrhythmias and atrial fibrillation (AF). Retrospective analysis of 3239 records from patients taking SGLT2 inhibitors vs 10405 taking other glucose lowering agents led to a significant reduction in “total” cardiac arrhythmias and AF.

General criticism:

1. The sample sizes are very impressive with a total of 62704 records screened. The resulting retrospective cohorts are still remarkably large with n=3239 in group 1 and 10405 in group 2. However, considering that globally about 75-85% of patients with T2DM are treated with oral glucose lowering drugs (GLDR), it requires further explanation, why 63% (39301 records) of the initially screened patients were excluded because they “did not use glucose-lowering agents” (p.3 study protocol chart). If only 37% of all T2DM patients in this cohort had been on GLDR, that cohort would hardly be representative for contemporary T2DM therapy. Please clarify.

2. Throughout the manuscript, the term “enrolled” is being used. Clinically and in legal terms, “Enrollment” is precisely defined, as in Clinicaltrials.gov as a reference site for clinical studies:

"Enrolled" is defined in 42 CFR 11.10(a) as a human subject's, or their legally authorized representative's, agreement to participate in a clinical trial following completion of the informed consent process, as required in 21 CFR Part 50 and/or 45 CFR Part 46, as applicable. (Clinicaltrials.gov)

Therefore, technically, „Enrollment“ requires a human subject that has been screened, meeting all inclusion/exclusion criteria, has signed an informed consent form and has received a study subject number. Therefore, the authors should add to “study population” a section detailing patient recruitment and informed consent, if applicable. If the data have been collected by retrospective screening of patient records only, the term “Enrollment”/ ”enrolled” does not apply and should be replaced with “included”/”allocated”/”registered” etc. Please specify.

3. Despite the fact that the onset or arrhythmia is a key outcome measure for the study, the manuscript remains rather vague with regards to any details: Please specify:

            1) What set of criteria was used to detect arrhythmia in the patient records?

            2) What criteria were used to distinguish AF from other arrhythmias

            3) What criteria were used to distinguish ventricular from atrial arrhythmias?

            4) What is meant by the frequently used term “total cardiac arrhythmias”

(e.g. p8, last paragraph)

            5) What drugs are included in “antiarrhythmia medication use” (p2, last paragraph). Have patients with betablocker therapy also been excluded?

4. The patient cohort should be characterized more precisely:

1)         Group 1 and 2 seem to differ significantly in all but one criterion studied (p5, Table 1), which is very remarkable and should lead to suspect selection bias. This disparity of study groups constitutes a relevant bias for further analysis. When calculating hazard ratios with Cox regression models, these have to be adjusted (adjusted hazard ratios) for any significant confounding variable between both groups, otherwise even a staged procedure of univariate and multivariate analysis will only produce biased values. It appears doubtful that this serious incongruence between both study groups could be counter-balanced by any statistical method.

One alternative might be to perform matched-pairs comparison in order to produce more comparable samples.

2)         Furthermore, it should be added as a limitation, that a number of those significant differences between group 1 and 2 are risk factors for atrial fibrillation by themselves (age, hypertension, etc) and present a relevant bias for the findings.

3)         Since about 80% of patients with T2DM globally are obese, the patients weight/BMI should be added.

4)         Since poor glycemic control is a risk factor for AF, MI, stroke etc, HbA1c as one of the most important markers for glycemic control should be added.

5)         Other crucial patient data, important not only as RF for AF, but also for heart failure, MI and other clinical conditions should be added: What about Coronary artery disease/ hx of MI, valvular heart disease (esp. mitral valve disorders), renal failure/GFR, Heart failure/diastolic dysfunction/LVEF/Left atrial size, Smoking status?

6)         Was thyroid function as a RF for AF also assessed?

7)         Did the authors assess hx of pacemaker implant?

8)         What was the motivation to include “gout” at such a prominent position and how was it assessed? Did the authors by any chance mean “goitre” instead of “gout”?

9)         Based on Table 1, zero % (0%) of patients in group 1 and 0.4% in group 2 had arterial hypertension. Is that credible with regards to a global prevalence for arterial hypertension of about 60-80% at ages 64-75 years? Did the authors mean “controlled” hypertension, which still would be absolutely unusual.

10)       Considering that GLDR are the main focus of the study, the percentage of substances prescribed in the non-SGLT2 group should be specified, e.g. what percentage of Metformin, what percentage of sulfonylurea etc. This also applies to the mention of “other glucose lowering agents” on P9, last paragraph.

5. Since the authors mention the term “real-world evidence”, it should be noted that retrospective studies are very well suited for rare conditions and are generally useful to create large cohort sizes with small error margins. However, retrospective analyses have specific downsides, including selection bias. (See: Collins et al., NEJM 2020, 382; 674-678: “The magic of randomization versus the myth of real-world evidence”).

Therefore, retrospective studies must be acknowledged as the inferior study design type compared to randomized trials and the absence of randomization in the context of the remarkably incongruent population samples in group 1 and group 2 of the current study should be discussed as a significant limitation rather than an asset. The authors may want to re-consider if it is wise to emphasize the idea of "real-world evidence" in such a prominent fashion.

6. In the discussion (p11, ll. 338 and 348) a blood pressure effect of SGLT2 inhibitors is being mentioned. The authors should add BP data in order to verify, if their cohort also shows BP changes depending on SGLT2 inhibitor therapy. Similarly, LVEF data would be interesting.

Specific criticism:

1. The p value as a marker of statistic significance should be printed small in the entire manuscript.

2. The language quality is rather poor throughout the manuscript. As a general rule, in English definite articles should be used with caution and are superfluous in about 25% of cases in the current manuscript (Example: P6, last paragraph: “The Group 1 patients had a lower incidence (…)”.

Generally speaking, there is a 95% probability that the term “to elucidate” is out of place when used in English medical literature (please use “show”, “demonstrate”, “clarify”, “confirm”, etc).

The entire manuscript needs to be cross-checked by an experienced English native speaker in order to improve comprehensibility.

3. Figure 2: If percentages (%) are shown at the y axis legend, which is the correct position, they should be omitted at the y axis tics labels.

4. On P10 the discussion of “Association between SGLT2 inhibitor therapy and AF” mentions only 3 RF for AF, “visceral adipose tissue, autonomic dysfunction and atrial fibrosis”. There is a large number of RF for AF with grave importance for clinical practice (obesity, CAD, MI, HF, mitral valve disease, LA size, hypertension, etc), which do not appear in the study cohort or in the discussion. Please substantiate your data.

Author Response

Reviewer 1

The authors present a retrospective cohort study on the association between SGLT2 therapy in patients with T2DM and onset of cardiac arrhythmias and atrial fibrillation (AF). Retrospective analysis of 3239 records from patients taking SGLT2 inhibitors vs 10405 taking other glucose lowering agents led to a significant reduction in “total” cardiac arrhythmias and AF.

General criticism:

  1. The sample sizes are very impressive with a total of 62704 records screened. The resulting retrospective cohorts are still remarkably large with n=3239 in group 1 and 10405 in group 2. However, considering that globally about 75-85% of patients with T2DM are treated with oral glucose lowering drugs (GLDR), it requires further explanation, why 63% (39301 records) of the initially screened patients were excluded because they “did not use glucose-lowering agents” (p.3 study protocol chart). If only 37% of all T2DM patients in this cohort had been on GLDR, that cohort would hardly be representative for contemporary T2DM therapy. Please clarify.

Reply: Thanks for your valuable opinion very much. We initially included the target group of patients as T2DM, however, some patients did not receive two or more DM diagnosis records for at least 3 months, so there is no record of using glucose-lowering agents. We had revised and add more description in our method. Please see our revised manuscript.

  1. Throughout the manuscript, the term “enrolled” is being used. Clinically and in legal terms, “Enrollment” is precisely defined, as in Clinicaltrials.gov as a reference site for clinical studies:

"Enrolled" is defined in 42 CFR 11.10(a) as a human subject's, or their legally authorized representative's, agreement to participate in a clinical trial following completion of the informed consent process, as required in 21 CFR Part 50 and/or 45 CFR Part 46, as applicable. (Clinicaltrials.gov)

Therefore, technically, „Enrollment“ requires a human subject that has been screened, meeting all inclusion/exclusion criteria, has signed an informed consent form and has received a study subject number. Therefore, the authors should add to “study population” a section detailing patient recruitment and informed consent, if applicable. If the data have been collected by retrospective screening of patient records only, the term “Enrollment”/”enrolled” does not apply and should be replaced with “included”/”allocated”/”registered” etc. Please specify.
Reply: Thanks for the reminder very much, the words of “Enrollment" and "enrolled" have been replaced with included to better results interpretation in the experimental design between groups.

  1. Despite the fact that the onset or arrhythmia is a key outcome measure for the study, the manuscript remains rather vague with regards to any details: Please specify:

            1) What set of criteria was used to detect arrhythmia in the patient records?

            2) What criteria were used to distinguish AF from other arrhythmias

            3) What criteria were used to distinguish ventricular from atrial arrhythmias?

            4) What is meant by the frequently used term “total cardiac arrhythmias”

(e.g. p8, last paragraph)

Reply: Appreciate your valuable opinion. The judgment on diseases such as arrhythmia and AF is based on the diagnosis code of ICD9/10 patients received in the database during 2016-2019. The diagnostic codes for Arrhythmia and AF are I49.9-AAPC and I48-AAPC, respectively. We had added in our method. Please see our revised manuscript.

5) What drugs are included in “anti-arrhythmia medication use” (p2, last paragraph). Have patients with betablocker therapy also been excluded?
Reply: We thanks your opinion very much. This study is a retrospective analysis of the KMUH database. The anti-arrhythmia drugs used in this study including dronedarone, propafenone, amiodarone, flecainide and mexiletine. Beta-blocker is not included. We had added more description in our revised manuscript. Please see our revised manuscript.

  1. The patient cohort should be characterized more precisely:

1)         Group 1 and 2 seem to differ significantly in all but one criterion studied (p5, Table 1), which is very remarkable and should lead to suspect selection bias. This disparity of study groups constitutes a relevant bias for further analysis. When calculating hazard ratios with Cox regression models, these have to be adjusted (adjusted hazard ratios) for any significant confounding variable between both groups, otherwise even a staged procedure of univariate and multivariate analysis will only produce biased values. It appears doubtful that this serious incongruence between both study groups could be counter-balanced by any statistical method.

One alternative might be to perform matched-pairs comparison in order to produce more comparable samples.

Reply: Thanks for the reminder. We have performed matched-pairs statistical analysis on the included data. The age- and sex-matched multivariate results showed that patients with SGLT2 inhibitors exhibited significant higher cardiac diseases/syndrome-free incidence than those patients with non-SGLT2 inhibitors. 

2)         Furthermore, it should be added as a limitation, that a number of those significant differences between group 1 and 2 are risk factors for atrial fibrillation by themselves (age, hypertension, etc) and present a relevant bias for the findings.

Reply: Thanks for your valuable opinions. The related limitation has been added in our revised manuscript.

3)         Since about 80% of patients with T2DM globally are obese, the patients weight/BMI should be added.

Reply: Appreciate your opinion. We're sorry regarding the incomplete values of weight/BMI in our database. We are working to optimize the construction of the database to more accurately predict the effects of drugs.

4)         Since poor glycemic control is a risk factor for AF, MI, stroke etc, HbA1c as one of the most important markers for glycemic control should be added.

Reply: Thanks for your valuable opinion very much. Statistics for HBA1c have been added (from table 1 to 6) in our revised manuscript.

5)         Other crucial patient data, important not only as RF for AF, but also for heart failure, MI and other clinical conditions should be added: What about Coronary artery disease/ hx of MI, valvular heart disease (esp. mitral valve disorders), renal failure/GFR, Heart failure/diastolic dysfunction/LVEF/Left atrial size, Smoking status? 

Reply: Appreciate your valuable opinions. These are very interesting topics. In our research design, the primary outcome such as arrhythmia and AF is something we are more concerned about. In this research, stroke, MI, and heart failure belong to the secondary outcome. We also consider continuing analyze the RF about heart failure, MI and stroke.

6)         Was thyroid function as a RF for AF also assessed?

Reply: Thanks for your opinions. We did not have the routine examination for thyroid function in our clinical DM care. Therefore, we did not have enough data to make statistics in our databank.

7)         Did the authors assess hx of pacemaker implant?   

Reply: Thanks for your valuable opinions. After we searched according to the diagnosis code “2022 ICD-10-CM Diagnosis Code Z95.0”, we found no relevant records in all the included patient data.

8)         What was the motivation to include “gout” at such a prominent position and how was it assessed? Did the authors by any chance mean “goitre” instead of “gout”? 

Reply: Thanks for your valuable opinion. Hyperuricemia and gout were believed risk factors for cardiac arrhythmia, especially atrial fibrillation, in previous report. Please see the reference below. That is why we include gout as a risk factor in our study.

Reference:

1.     YJ Kuo, TH Tsai and HP Chang, et al. The risk of atrial fibrillation in patients with gout: a nationwide population-based study. Sci Rep 2016; 6: 32220

9)         Based on Table 1, zero % (0%) of patients in group 1 and 0.4% in group 2 had arterial hypertension. Is that credible with regards to a global prevalence for arterial hypertension of about 60-80% at ages 64-75 years? Did the authors mean “controlled” hypertension, which still would be absolutely unusual. 

Reply: Appreciated your valuable opinions very much. We've re-checked that in our total included patient data, the average percentage of hypertension was 47%. However, our exclusion criteria excluded patients diagnosed with hypertension after medication administration, which almost all of the hypertension patients were ruled out. Thus, to more accurately describe our investigation, the title of the current study had been changed to “Clinical Observation of SGLT2 Inhibitor Therapy for Cardiac Arrhythmia and Related Cardiovascular Disease in Non-hypertensive Diabetic Patients”

10)       Considering that GLDR are the main focus of the study, the percentage of substances prescribed in the non-SGLT2 group should be specified, e.g. what percentage of Metformin, what percentage of sulfonylurea etc. This also applies to the mention of “other glucose lowering agents” on P9, last paragraph.

Reply: Appreciated your valuable opinions. We further analyzed the use of non-SGLT2 inhibitors based on drug codes. The analysis results are shown in the table below. The most commonly used drugs were DPP 4 inhibitors, which were used in 87% of the final matched patients with non-SGLT2 inhibitors, followed with 40% Sulfonylureas accounted, 23% Thiazolidinediones, and 5.9 % Repaglinide. Clinically, metformin is common used in combination therapy with other glucose-lowering agents such as Sulfonylureas, Thiazolidinediones, DPP 4 inhibitors drugs.

non-SGLT2, N = 6,406

Metformin

3,587/6,406 (56%)

Sulfonylureas

2,581 / 6,406 (40%)

Thiazolidinediones

1,498 / 6,406 (23%)

Repaglinide

377 / 6,406 (5.9%)

DPP 4 inhibitors

5,568 / 6,406 (87%)

  1. Since the authors mention the term “real-world evidence”, it should be noted that retrospective studies are very well suited for rare conditions and are generally useful to create large cohort sizes with small error margins. However, retrospective analyses have specific downsides, including selection bias. (See: Collins et al., NEJM 2020, 382; 674-678: “The magic of randomization versus the myth of real-world evidence”).

Therefore, retrospective studies must be acknowledged as the inferior study design type compared to randomized trials and the absence of randomization in the context of the remarkably incongruent population samples in group 1 and group 2 of the current study should be discussed as a significant limitation rather than an asset. The authors may want to re-consider if it is wise to emphasize the idea of "real-world evidence" in such a prominent fashion.

Reply: Appreciated your opinions very much. Considering the strict definition of real-world evidence, our research has been changed to "clinical practice evidence" to more in line with our research design and results.

  1. In the discussion (p11, ll. 338 and 348) a blood pressure effect of SGLT2 inhibitors is being mentioned. The authors should add BP data in order to verify, if their cohort also shows BP changes depending on SGLT2 inhibitor therapy. Similarly, LVEF data would be interesting.

Reply: Thanks for your valuable opinions. In our databank, there were missing medical records, include BP and LVEF data, to perform statistics. These were the limitations in our investigation.

Specific criticism:

  1. The p value as a marker of statistic significance should be printed small in the entire manuscript.

Reply: Thanks for the valuable reminder. These had been corrected. Please see our revised manuscript

  1. The language quality is rather poor throughout the manuscript. As a general rule, in English definite articles should be used with caution and are superfluous in about 25% of cases in the current manuscript (Example: P6, last paragraph: “The Group 1 patients had a lower incidence (…)”.

Reply: Appreciate your valuable opinions. This manuscript has been re-edited in English thoroughly.

Generally speaking, there is a 95% probability that the term “to elucidate” is out of place when used in English medical literature (please use “show”, “demonstrate”, “clarify”, “confirm”, etc).

Reply: Thanks for your valuable opinions. These had been corrected. Please see our revised manuiscript

The entire manuscript needs to be cross-checked by an experienced English native speaker in order to improve comprehensibility.

Reply: Thanks for your valuable opinions very much. This manuscript has been English re-edited thoroughly.

  1. Figure 2: If percentages (%) are shown at the y axis legend, which is the correct position, they should be omitted at the y axis tics labels.

Reply: Thanks for your valuable opinions. These had been corrected.

  1. On P10 the discussion of “Association between SGLT2 inhibitor therapy and AF” mentions only 3 RF for AF, “visceral adipose tissue, autonomic dysfunction and atrial fibrosis”. There is a large number of RF for AF with grave importance for clinical practice (obesity, CAD, MI, HF, mitral valve disease, LA size, hypertension, etc), which do not appear in the study cohort or in the discussion. Please substantiate your data.

Reply: Thank you for your suggestions. In the data we obtained from database, those RF are not included.

Reviewer 2 Report

1)Abstract.

What is group 1and group 2-  explanation is needed.

2) Material and methods.

The inclusion and exclusion criteria should be described more precisely.

3) Did authors obtained the Bioethics Committee  Approval?

4) Table 1.The hypertension was diagnosed in 0%  of  group 1 and 0.4 % of group 2. It is unexpected, because hypertension and diabetes  are two diseases that frequently co-exist in the general population, even in 50%. This result should be discussed.

What was BMI of studied groups?

5) 166  The next unexpected results is that Total cholesterol corresponded to decresed incidence of cardiac arrythmia . Why? Disscusion is needed.

6) 186  Morover , Total colesterol was negatively coreelated with incidence of atrial fibrillation. T-cholestorol  strong risk factor for atherosclerosis, which is one of the most common cause of AF .Therefore, this results is  very unexpected.

7)  220 The LDL  reduce risk for heart failure. This result is contrary to modern medical science and omitted in discussion section.

8) The most important result is reduced risk of cardiac arrythmia in SGLT2 treated patients. Could authors showed NNT (numer needed to treat to avoid 1 incidence of arrythmia?) It would be usuful for practitioners.

Author Response

 Reviewer 2

1)Abstract.

What is group 1and group 2-  explanation is needed.

Reply: Thanks for your valuable opinions. More explanation has been added in our revised manuscript.

2) Material and methods.

The inclusion and exclusion criteria should be described more precisely.

Reply: Appreciate your valuable opinions. More explanations and definition about inclusion and exclusion have been added in the Study population, the 1st paragraph of Material and Methods section

3) Did authors obtained the Bioethics Committee Approval?    

Reply: Thanks for your valuable opinions. Yes, it has been uploaded in our revised manuscript.

4) Table 1. The hypertension was diagnosed in 0% of group 1 and 0.4 % of group 2. It is unexpected, because hypertension and diabetes are two diseases that frequently co-exist in the general population, even in 50%. This result should be discussed.

Reply: Appreciate your valuable opinions very much. We've re-checked that in our total included patient data, the average percentage of hypertension was 47%. However, our exclusion criteria excluded patients diagnosed with hypertension after medication administration, which all of the uncontrolled hypertension patients were ruled out. Thus, to more accurately describe our investigation, the title of the current study had been changed to “Clinical Observation of SGLT2 Inhibitor Therapy for Cardiac Arrhythmia and Related Cardiovascular Disease in Non-hypertensive Diabetic Patients”

What was BMI of studied groups?

Reply: Thanks for your valuable opinions. There is no BMI value in the database, so we can't count it for now. We are working to optimize the construction of the database to more accurately predict the effects of drugs.

5) The next unexpected results is that Total cholesterol corresponded to decresed incidence of cardiac arrythmia. Why? Disscusion is needed.

Reply: Thanks for your valuable opinions. In our initial investigation, we did not perform match-paired analysis between group 1 and group 2. Therefore, there was bias in our study. We had re-performed match-pair analysis and total cholesterol was not corresponded to decrease the incidence of cardiac arrhythmia. Please see our revised manuscript.

6) 186  Morover , Total colesterol was negatively coreelated with incidence of atrial fibrillation. T-cholestorol strong risk factor for atherosclerosis, which is one of the most common cause of AF. Therefore, this results is very unexpected.

Reply: Appreciate your valuable opinions. In our investigation, the total cholesterol level was a significantly decreasing the incidence of atrial fibrillation. “Cholesterol paradox in atrial fibrillation” was also extensive discussed in previous reports. Cholesterol is a main component of the cell membrane, and changes in cholesterol levels can cause changes in membrane properties through effects on membrane permeability and membrane proteins such as ion channels, pumps, and receptors. This can affect electrical gradient and resting potential across the membranes and potentiate the development of arrhythmias. Low levels of cholesterols can also reflect the level of inflammation which might contribute to cardiac arrhythmia. In addition, old age or hyperthyroidism is associated with low cholesterol levels, and increased incidence of AF. Please see the reference below.

Reference

  1. HJ Lee, SR Lee, EK Choi, et al. Low Lipid Levels and High Variability are Associated With the Risk of New-Onset Atrial Fibrillation. J Am Heart Assoc 2019; 8(23): e012771

7)  220 The LDL reduce risk for heart failure. This result is contrary to modern medical science and omitted in discussion section.

Reply: Thanks for your valuable opinions. We revalidated our analytical model. We conducted an age- and sex-matched analysis with group 1: SGLT2 inhibitors users, and found that there was no significant difference in LDL-C, T-COL in analysis of arrhythmia, stroke, heart failure and MI in groups 1 and 2. They are confounding factors in many cardiac diseases.

8) The most important result is reduced risk of cardiac arrythmia in SGLT2 treated patients. Could authors showed NNT (number needed to treat to avoid 1 incidence of arrythmia?) It would be usuful for practitioners.

Response: Appreciate your valuable opinions very much. The NNT for arrhythmia is 166.66, which means that 1 in 166.66 patients on SGLT2 inhibitors can avoid arrhythmia.

Round 2

Reviewer 1 Report

General criticism

The authors have invested substantial effort to address the points of criticism put forward in the last review. This applies especially to the peculiarities of the study population, frequency of arterial hypertension and additional background variables as classification of arrhythmias and HbA1C values. Considering the incongruent nature of the

  1. Hypertension

It is a reasonable solution to classify the patients as “controlled hypertension” if they all had normal blood pressure values under antihypertensive therapy. With regards to the fact that this is only an indicator of the question to what extent these patient data can be accepted as representative for everyday patients, I would not insist on this detail to be included into the study title (as has been done in the revision), since this criterion has been adequately highlighted in the abstract and the “Materials and methods” section. Furthermore, strictly speaking, the patients are not “Non-hypertensive”, if BP was well controlled under antihypertensive therapy. As stated correctly in the text, they would have to be classified as “controlled hypertension” patients.

I would leave it to the authors preference if they prefer change the title to “…and related cardiovascular disease in diabetic patients with controlled hypertension” or (which would be my recommendation) to leave it out altogether.

  1. Statistics

Case-based comparisons are a valuable addition to the statistics. It is reassuring that case-based comparisons are still producing significant differences consistent with the studies hypothesis. Unfortunately, due to the “track-changes” version of the documents with the correction record obstructing the right part of the page, the far right column in almost all tables cannot be read properly. Also, due to the space taken up by the correction inserts on the right, the layout of all tables seems to be compromised. I would need either a word document of the manuscript or a pdf version with “track-changes” all accepted to assess the very right column of the tables.

Minor criticism

Still some spelling and grammar issues:

  1. p2 l32: “The secondary study outcomes…” -> “Secondary study outcomes were…”
  2. p2 l40: “The multivariate analysis showed…” -> “Multivariate analysis showed…”
  3. p3 l94: “…age less than 18-year-old,…” -> “… age less than 18 years,…”
  4. p3 l96: “glucose lowering agent” -> “glucose lowering agents”
  5. p4 l114: “The primary study outcome…” -> “The primary study outcome was the incidence of atrial and ventricular arrhythmias and their association with SGLT2 inhibitor therapy. The secondary study outcome was a compound of related cardiovascular events including stroke, heart failure and myocardial infarction and their association with SGLT2 inhibitor therapy. “
  6. p4 l125: “Fisher’s exact tests” -> “Fisher’s exact test”
  7. p8 l167: “… there were no significance between age, sex and GPT…” -> “… there were no significant differences between age, sex and GPT…”
  8. p8 l170 “Generally the age of included patients were in the average of 65 year old, with the 58-59% male.” -> “The average age of included patients was about 65 years with 58-59% being male”
  9. p10 l187: “incidence (…) after SGLT2 therapy…” -> “incidence (…) in the presence of SGLT2 therapy”

Author Response

Reviewer 1

Comments and Suggestions for Authors

General criticism

The authors have invested substantial effort to address the points of criticism put forward in the last review. This applies especially to the peculiarities of the study population, frequency of arterial hypertension and additional background variables as classification of arrhythmias and HbA1C values. Considering the incongruent nature of the

Hypertension

It is a reasonable solution to classify the patients as “controlled hypertension” if they all had normal blood pressure values under antihypertensive therapy. With regards to the fact that this is only an indicator of the question to what extent these patient data can be accepted as representative for everyday patients, I would not insist on this detail to be included into the study title (as has been done in the revision), since this criterion has been adequately highlighted in the abstract and the “Materials and methods” section. Furthermore, strictly speaking, the patients are not “Non-hypertensive”, if BP was well controlled under antihypertensive therapy. As stated correctly in the text, they would have to be classified as “controlled hypertension” patients.

I would leave it to the authors preference if they prefer change the title to “…and related cardiovascular disease in diabetic patients with controlled hypertension” or (which would be my recommendation) to leave it out altogether.

Reply: Thanks for the suggestion. We also agree that the inclusion of normotensive patients may be due to the contribution of antihypertensive drugs with a history of hypertension. This does not mean that high blood pressure is completely cured. In this case, "controlled hypertension" is more appropriate than "non-hypertensive".

Statistics

Case-based comparisons are a valuable addition to the statistics. It is reassuring that case-based comparisons are still producing significant differences consistent with the studies hypothesis. Unfortunately, due to the “track-changes” version of the documents with the correction record obstructing the right part of the page, the far right column in almost all tables cannot be read properly. Also, due to the space taken up by the correction inserts on the right, the layout of all tables seems to be compromised. I would need either a word document of the manuscript or a pdf version with “track-changes” all accepted to assess the very right column of the tables.

Reply: Our original revised manuscript had corrections highlighted in red, but probably due to the automatic comparison of the MDPI submission system, adding "track-changes" ended up compromising the columns of the table. The revised Tables 1-6 are attached below for your convenience.

Table 1. Characteristics of the Patients Included in the Study.

Characteristic (mean (SD))

Group 1                               SGLT2, N = 3,203

Group 2                                     non-SGLT2, N = 6,406

P value

Age, years

65.333 (12.128)

65.628 (11.712)

0.6

Male sex, n (%)

1,867/3,203 (58%)

3,757/6,406 (59%)

0.7

GPT

28.151 (15.982)

29.808 (21.543)

0.2

HbA1c

7.513 (1.217)

7.139 (1.178)

<0.001

LDL-C

88.800 (26.372)

92.003 (26.190)

<0.001

T-CHO

158.809 (33.749)

163.341 (33.105)

<0.001

GLU (AC)

142.228 (36.529)

140.211 (36.173)

0.007

HDL-C

44.694 (11.512)

43.298 (11.648)

<0.001

TG

145.235 (132.144)

145.690 (100.912)

<0.001

Gout

2/3,203 (<0.1%)

13/6,406 (0.2%)

0.2

Table 2. Univariate and Multivariate Analyses of Total Cardiac Arrhythmia Incidence in the Study Patients.

Variable

Univariate analysis

Multivariate analysis

Matched-pairs analysis

HR

95% CI

p-value

HR

95% CI

p-value

HR

95% CI

p-value

SGLT2 inhibitors therapy

0.5

(0.33-0.74)

0.001

0.61

(0.41-0.92)

0.019

0.58

(0.38-0.89)

0.013

Age

2.3

(1.9-2.7)

<0.001

1.95

(1.64-2.31)

<0.001

Male

1.1

(0.85-1.5)

0.450

GPT

0.74

(0.61-0.91)

0.005

LDL-C

0.81

(0.7-0.94)

0.005

1.64

(1.20-2.26)

0.002

T-CHO

0.7

(0.6-0.82)

<0.001

0.54

(0.38-0.77)

<0.001

GLU(AC)

0.92

(0.8-1.1)

0.270

HDL-C

0.71

(0.61-0.83)

<0.001

0.73

(0.58-0.91)

0.005

Gout

5.9

(1.5-24)

0.012

HbA1C

1

(0.87-1.2)

0.66

Table 3. Univariate and Multivariate Analyses of Atrial Fibrillation Incidence in the Study Patients.

Variable

Univariate analysis

Multivariate analysis

Matched-pairs analysis

HR

95% CI

p-value

HR

95% CI

p-value

HR

95% CI

p-value

SGLT2 inhibitors therapy

0.48

(0.31-0.73)

0.001

0.6

(0.39-0.94)

0.025

0.56

(0.35-0.88)

0.013

Age

2.4

(2.1-2.9)

<0.001

2.05

(1.71-2.47)

<0.001

Male

1

(0.76-1.4)

0.9

GPT

0.64

(0.5-0.82)

<0.001

LDL-C

0.79

(0.68-0.93)

0.004

1.59

(1.13-2.24)

0.008

T-CHO

0.7

(0.59-0.82)

<0.0001

0.55

(0.38-0.80)

0.002

0.64

(0.41-1.00)

0.048

GLU(AC)

0.89

(0.76-1)

0.150

HDL-C

0.74

(0.63-0.87)

<0.001

Gout

6.7

(1.7-27)

0.008

HbA1C

1.1

(0.88-1.3)

0.55

Table 4. Univariate and Multivariate Analyses for Stroke Incidence in the Study Patients.

Variable

Univariate analysis

Multivariate analysis

Matched-pairs analysis

HR

95% CI

p-value

HR

95% CI

p-value

HR

95% CI

p-value

SGLT2 inhibitors therapy

0.41

(0.29-0.59)

<0.0001

0.51

(0.35-0.73)

<0.001

0.48

(0.33-0.7)

<0.001

Age

1.6

(1.4-1.8)

<0.0001

1.3

(1.14-1.47)

<0.001

Male

1.1

(0.88-1.4)

0.39

GPT

0.9

(0.78-1)

0.13

LDL-C

1.1

(0.96-1.2)

0.23

T-CHO

1

(0.93-1.2)

0.46

GLU(AC)

1.1

(0.95-1.2)

0.31

HDL-C

0.66

(0.58-0.75)

<0.0001

0.72

(0.63-0.82)

<0.001

0.61

(0.51-0.74)

<0.001

Gout

4.2

(1.1-17)

0.042

HbA1C

1

(0.87-1.2)

0.86

Table 5. Univariate and Multivariate Analyses of Heart Failure Incidence in the Study Patients.

Variable

Univariate analysis

Multivariate analysis

Matched-pairs analysis

HR

95% CI

p-value

HR

95% CI

p-value

HR

95% CI

p-value

SGLT2 inhibitors therapy

0.51

(0.4-0.64)

<0.001

0.6

(0.47-0.77)

<0.001

0.54

(0.41-0.7)

<0.001

Age

1.9

(1.7-2.1)

<0.001

1.57

(1.41-1.74)

<0.001

Male

0.85

(0.72-1)

0.063

GPT

66

(0.57-0.76)

<0.001

0.82

(0.72-0.94)

0.003

LDL-C

0.78

(0.71-0.85)

<0.001

0.79

(0.68-0.92)

0.002

T-CHO

0.85

(0.77-0.93)

<0.001

1.24

(1.08-1.43)

0.003

GLU(AC)

0.99

(0.91-1.1)

0.86

HDL-C

0.68

(0.62-0.75)

<0.001

0.77

(0.70-0.84)

<0.001

0.78

(0.68-0.89)

<0.001

Gout

8.6

(4.1-18)

<0.001

HbA1C

1.1

(1-1.3)

0.019

1.18

(1.07-1.31)

0.001

Table 6. Univariate and Multivariate Analyses of Myocardial Infarction Incidence in the Study Patients.

Variable

Univariate analysis

Multivariate analysis

Matched-pairs analysis

HR

95% CI

p-value

HR

95% CI

p-value

HR

95% CI

p-value

SGLT2 inhibitors therapy

0.5

(0.34-0.74)

<0.001

0.47

0.31-0.70

<0.001

0.47

(0.31-0.72)

<0.001

Age

1.7

(1.5-1.9)

<0.001

1.46

1.23-1.72

<0.001

Male

1.5

(1.2-2)

0.0025

1.68

1.24-2.26

<0.001

GPT

0.66

(0.53-0.83)

<0.001

0.76

0.61, 0.95

0.018

LDL-C

0.62

(0.53-0.73)

<0.001

T-CHO

0.64

(0.55-0.74)

<0.001

GLU(AC)

0.96

(0.84-1.1)

0.58

HDL-C

0.56

(0.48-0.66)

<0.001

0.71

0.60, 0.85

<0.001

0.63

(0.5-0.79)

<0.001

Gout

<0.0001

(0-Inf)

0.99

HbA1C

1.2

(0.98-1.4)

0.077

Minor criticism

Still some spelling and grammar issues:

p2 l32: “The secondary study outcomes…” -> “Secondary study outcomes were…”

p2 l40: “The multivariate analysis showed…” -> “Multivariate analysis showed…”

p3 l94: “…age less than 18-year-old,…” -> “… age less than 18 years,…”

p3 l96: “glucose lowering agent” -> “glucose lowering agents”

p4 l114: “The primary study outcome…” -> “The primary study outcome was the incidence of atrial and ventricular arrhythmias and their association with SGLT2 inhibitor therapy. The secondary study outcome was a compound of related cardiovascular events including stroke, heart failure and myocardial infarction and their association with SGLT2 inhibitor therapy. “

p4 l125: “Fisher’s exact tests” -> “Fisher’s exact test”

p8 l167: “… there were no significance between age, sex and GPT…” -> “… there were no significant differences between age, sex and GPT…”

p8 l170 “Generally the age of included patients were in the average of 65 year old, with the 58-59% male.” -> “The average age of included patients was about 65 years with 58-59% being male”

p10 l187: “incidence (…) after SGLT2 therapy…” -> “incidence (…) in the presence of SGLT2 therapy”

Reply: Thanks for all suggestions and corrections.

Reviewer 2 Report

The authors revised the manuscript according to the reviewer's recommendations. 

Author Response

Reply: Thanks for all suggestions and corrections.